# Increased Progesterone on the Day of Administration of hCG in Controlled Ovarian Hyperstimulation Affects the Expression of HOXA10 in Primates’ Endometrial Receptivity

**DOI:** 10.3390/biomedicines7040083

**Published:** 2019-10-21

**Authors:** Nurhuda Sahar, Ninik Mujihartini, Dwi Ari Pudjianto, Adhea Debby Pradhita, Rosalina Thuffi, Kusmardi Kusmardi

**Affiliations:** 1Depatment of Medical Biology, Faculty of Medicine, Universitas Indonesia, Jakarta 10430, Indonesia; nurhuda.ms@ui.ac.id (N.S.); dwi.ari@ui.ac.id (D.A.P.); 2Department of Biochemistry and Molecular Biology, Faculty of Medicine, Universitas Indonesia, Jakarta 10430, Indonesia; ninikbiokim@yahoo.com; 3Biomedical Science Master Program, Faculty of Medicine, Universitas Indonesia, Jakarta 10430, Indonesia; andheadebbypradhita@gmail.com (A.D.P.); ochathuffi@gmail.com (R.T.); 4Department of Pathology Anatomy, Faculty of Medicine, Universitas Indonesia, Jakarta 10430, Indonesia

**Keywords:** progesterone, hCG administration, endometrial receptivity, COH, HOXA10

## Abstract

The increase in progesterone (P4) levels on the day of human chorionic gonadotropin (hCG) administration have a negative effect on endometrial receptivity. There are few reports regarding the expression of homeobox A10 (HOXA10) as one of many biomolecular factors of endometrial receptivity. To evaluate the effect of increased P4 concentration on the day of hCG administration on HOXA10, a total of 16 *Macaca nemestrina* were divided into three dose groups of recombinant-follicle stimulating hormone (rFSH) (30IU, 50IU, and 70IU) and one control group. Injection of rFSH combined with gonadotropin release hormone (GnRH) at 160 ug/day was given subcutaneously using a long protocol technique. Blood samples for estradiol (E2) and (P4) concentration measurements were taken on the day of injecting hCG in the final follicular phase, while the collection of endometrial tissue for HOXA10 measurement was carried out 8 to 10 days after hCG administration. E2 and P4 were measured by ELISA, whereas HOXA10 expression was measured with immunohistochemical (IHC) techniques. The concentration of E2 and P4 was found to be higher in dose groups compared with the natural group, but no significant differences were found within the group. For the Hscore for HOXA10 expression, no significant differences within dose groups were found. In addition, no significant differences for the Hscore for HOXA10 were found when compared to E2 groups. Significantly, the Hscore of HOXA10 was found to be >1 ng/mL in the P4 group compared with the Hscore HOXA10 in the P4 natural group (*p* = 0.022). The high concentration of P4 caused by ovarian hyperstimulation in the follicular phase stimulates the expression of HOXA10 in the secretion phase.

## 1. Introduction

Controlled ovarian hyperstimulation (COH), a combination of recombinant-follicle stimulating hormone (rFSH) and gonadotropin release hormone (GnRH) agonist or antagonist protocol, is a standard procedure in In Vitro Fertilization (IVF) to stimulate the maturation of many ovarian follicles. Although there has been an increase in the number of embryos that have been produced, the implantation rate remains low, which is around 20% to 30% per fresh cycle [1]. Some studies have reported that COH may have a negative effect on endometrial receptivity [2]. Our hypothesis is the effect of COH on endometrial receptivity may occur through increased secretion of steroid hormones, especially progesterone.

The increase in progesterone levels at the end of follicular or on the day of hCG administration, in the COH procedure, reaches 38% [3]. The impact on pregnancy is still debated. Some researchers have reported that progesterone hypersecretion on the day of hCG administration negatively affects the pregnancy rate [4,5,6,7,8,9,10], whereas other researchers obtained different results [11]. A progesterone threshold value of more than 1.5 ng/mL on the day of hCG administration significantly reduces pregnancy rates compared to a progesterone level of <1.5 ng/mL, but the direct effect of the implantation period on endometrial receptivity has not been reported [10,11].

Homeobox A10 (HOXA10) is a transcription factor protein that regulates the development of the uterus and plays a role in endometrial development throughout the menstrual cycle [12]. Hoxa10 is a molecular marker of endometrial reception characteristics with peak expression shown during the implantation window [13,14,15]. Abnormal Hoxa10 expression is associated with endometrial reception disorders [16]. HOXA10 has been widely investigated as a marker to evaluate endometrial reception during the implantation period. In our study, we evaluated the effect of progesterone on the day of hCG administration with a threshold value of more than 1 ng/mL against HOXA10 expression as a marker of endometrial receptivity. This study aimed to compare HOXA10 expression levels in *Macaca nemestrina* endometrium between the progesterone (P4) > 1 ng/mL and P4 < 1 ng/mL in the stimulated cycle compared with P4 in the natural cycle.

## 2. Materials and Methods

### 2.1. Animal

The animals used in this experimental study were female (*Macaca nemestrina*) at reproductive age, 8 to 10 years, a weight of 5 to 8 kg, that had already given birth. The animals were obtained from the Primates Animal Study Center of the Bogor Agricultural Institute, Bogor, Indonesia. The study protocol was approved by the Institutional Animal Care and Use Committee for Primate Animal Studies of Bogor Agricultural Institute (ACUC No.08-B001-1R, February 01, 2016). This research was part of a bigger project where the majority of the organs of *Macaca nemestrina* were used. It provided the samples needed to execute the study.

*Macaca nemestrina* were chosen as the sample model because of the similarity in the reproductive system, for example, in the neuroendocrinology system, organ function, menstrual cycle, reproduction pathway, formation and control of mother–fetus–placenta during pregnancy, and reproductive aging from puberty to menopause. Mice are found to have different pathways of the reproduction system, in this case, is in the regulation of steroids [13,14,15]. Estradiol is a major regulator in endometrial mice, but not in humans or primates. The animals selected for use in this study were tattooed with an identification number and were housed in individual cages made of stainless material. All the animals were quarantined and adapted to new individual cages for two to three menstrual cycles. During this time, animal health was maintained, and any treatment was administered as needed.

### 2.2. Controlled Ovarian Hyperstimulation Procedure

For the controlled ovarian hyperstimulation (COH) procedure, a combination of gonadotrophin was administered with a long GnRH protocol using one of the three following regimens: 1. rFSH Gonal F (Merck KGaA, Darmstadt, Germany) in three dose groups (30IU, 50IU, and 70IU), 2. GnRH agonist (suprefact) (Sanofi S.A., Paris, France). 3. hCG (Pregnyl; Merck KGaA). The GnRH agonist was administered at a dose of 160 µg beginning in the luteal phase in the middle of the previous menstrual cycle and continued until the day before ovulation (approximately 14 days). After obtaining the E2 hormone level <70 pg/mL on the second day of menstruation, the administration was combined with rFSH at the dose of 30, 50, and 70 IU for the three treatment groups. rFSH was injected on the second day after menstruation at a dose according to the treatment group for 10 days until E2 secretion had peaked. Furthermore, hCG was administered at a dose of 10,000 IU or equivalent to 3200 IU. The luteal phase was determined by measuring serial P4 levels starting on the post ovulation day (Figure 1).

### 2.3. Endometrial Collection

The uterus of each animal was collected at 9 to 10 days after the peak of E2 secretion in the normal menstrual cycle group and the stimulated groups. Before surgery, each animal was anesthetized with ketamine at a dose of 0.1 mL/kg body weight. At necropsy, the uterus was rinsed with phosphate buffer (PBS), and a portion of tissue was incubated in a 10% formalin solution and then embedded in paraffin.

### 2.4. Hormone Assay

E2 serum on the day of the hCG administration was measured by a competitive immunoenzymatic assay. The sensitivity of the assay was 10 pg/mL, and the intra-assay coefficient of variation was 5%. P4 serum on the day of the hCG administration was determined by a competitive chemiluminescent immunoassay (IMMULITE, DPC, Los Angeles, CA, USA). The sensitivity of the method was 0.2 ng/mL, and the intra-assay coefficient of variation was 6.7%. Blood was allowed to clot, and serum was separated and stored at −20 °C until assayed.

### 2.5. Immunohistochemistry for HOXA10

In this study, immunohistochemical procedures were somewhat like the procedures performed by Rackow et al., 2008 [17]. Thin pieces paraffin sections were cut and mounted on poly-l-lysine-coated slides and allowed to air dry. Sections were deparaffinized in xylene, hydrated in descending grades of alcohol, and rinsed with water flow. Slides were incubated for 30 min in an ethanol solution containing 0.5% H_2_O_2_ and then washed in running water. Retrieval of the antigen HOXA10 was done by using the tool BioCare decloaking Chamber at a temperature of 120 to 125 °C for 5 min and then washed in 0.01 M PBS (pH 7,2). Slides were incubated for 20 min in 0.01 M PBS (pH 7.2) containing 3% normal horse serum (NHS) for the blocking antibody and then rinsed in PBS. Incubated and probed overnight with an optimized concentration (1:300) of the primary goat polyclonal HOXA10 antibody (Santa Cruz Biotechnology, Santa Cruz, CA, USA). Negative controls were incubated with normal goat serum instead of the primary antibody. The next day the slides were washed five times with PBS and incubated with donkey anti-goat biotinylated secondary antibody (Santa Cruz Bio-technology, CA, USA) for 2 h. After five washes for 5 min each, the sections were incubated for 30 min with streptavidin–HRP (Santa Cruz Biotechnology) and then washed again in PBS.

Chromogen diaminobenzidine (DAB, Biomedical BioCare USA) was added and allowed to stand for one minute, then washed in running water. After being washed and then stained by hematoxylin and closed with entelan, the resulting staining was evaluated by a light microscope at 100 times magnification objective, then photographed five times on the functional area. The Hscore was calculated using the following equation: Hscore = ΣPi (i + 1) where i is the intensity of staining with a value of 1, 2, and, 3 (weak, moderate, and strong, respectively) and Pi is the percentage of stained cells, varying from 0% to 100%.

### 2.6. Statistical Analysis

The measurement data shown are the average value with a standard deviation in normal data and median in not-normal data. *p* < 0.05 is considered significant, denoted by (*). Data were analyzed parametrically by ANOVA and non-parametrically by Kruskal–Wallis using SPSS 23 Software (IBM Corporation).

## 3. Result

### 3.1. Location of HOXA10 Expression in Macaca nemestrina Endometrium

The expression of HOXA10 protein in the endometrium mid-luteal phase of *Macaca nemestrina* was detected in the nucleus of the stromal cells and cytoplasm of glandular epithelial cells (Figure 2). Strong immunoreactivity was seen in the nucleus of stromal cells and decrease in the cytoplasm of glandular epithelial cells. These results were in line with those of Matsuzaki et al. 2009 [18] and Browne and Taylor, 2006 [19]. Abundant HOXA10 expression in the endometrial stromal cell nucleus shows its role not only in the embryonic implantation but also in the uterine decidualization [20,21].

### 3.2. Hormone Serum Progesterone and HOXA10 Hscore Levels in Natural Cycles and Stimulated Cycles

The average E2 and P4 level on the day of hCG administration were higher in the stimulated cycle group compared to the natural cycle group although the Kruskal–Wallis test showed no significant differences (*p* > 0.05) (Figure 2C,D). Likewise, the H-score value of HOXA10 expression in the stimulated cycle did not show a significant difference but with decreased pattern (Figure 2B).

Although there was no significant difference in E2 and P4 concentration based on the dose group, there were some patterns of concentration, a grouping correlation analysis with HOXA10 changes from dose group to concentration group. E2 concentration divided into three groups (control, E2 ≤ 1000, and E2 > 1000), and P4 concentration also divided into three groups (control, P4 ≤ 1, and P4 > 1).

The Kruskal–Wallis test showed no significant difference in the expression of HOXA10 within E2 concentration groups. A significant difference in HOXA10 expression was found within the P4 group, where group P4 > 1 had a lower expression of HOXA10 than the control group (*p* < 0.05) (Figure 3). The mean Hscore of HOXA10 expression in natural cycles as the control and cycles stimulated with P4 ≤ 1 ng/mL and P4 > 1 ng/mL, was 2.78 vs. 2.68 vs. 2.38, respectively. Figure 4 shows the different expression of the endometrium tissue within P4 groups.

## 4. Discussion

The decreased pregnancy rate in the cycle stimulated in IVF is a negative influence of the COH procedure on endometrial receptivity [4,5,6,7,8,9], but a specific mechanism cannot be explained. It is assumed that the effect of COH on endometrial receptivity occurs through an increase in steroid hormone levels. In this experiment, primate animals were used as objects, because in humans, it is difficult to get a sample of the uterus, constrained by ethical problems. The effect of P4 threshold value >1 ng/mL on the day of hCG administration on endometrial primate receptivity was analyzed through HOXA10 expression as a marker of endometrial receptivity in the implantation period.

The indicator of the success in COH procedures was there was more than one follicle that developed in one menstruation cycle. It could be observed by the concentration of the E2 secretion at the end of the follicular phase or on the day of hCG administration. Our data shows that 81% (9 samples) showed a positive response, where E2 and P4 levels were higher than the average in the natural cycle (data not shown). Animal response to the administration of ovarian stimulator regimens did not seem to be determined by dose level but affected by individuals, such as ovarian capacity and the number of antral follicles [22,23].

In this study, it showed that the use of recombinant FSH and those combined with GnRH agonist on COH did not directly affect HOXA10 expression. On the other hand, the direct impact of increasing P4 levels on the day of hCG administration has never been studied. Therefore, we further analyzed the decrease in endometrial receptivity based on the level of P4 on the day of hCG administration. Our data showed that the Hscore value of HOXA10 expression in the P4 > 1 ng/mL significantly decreased compared to the P4 ≤ 1 ng/mL and P4 level in the natural cycle as control. This showed that the P4 > 1 ng/mL on the day of hCG administration directly impaired the acceptability of endometrium in the implantation period by decreasing the expression of HOXA10 [24]. Miller et al., 2012, reported that the number of pregnancies and deliveries was higher in women with a normal level of receptive markers than low expressions [25]. Our data can explain the trends—the lower clinical pregnancy rate in the cycle was stimulated compared to the natural cycle. Our findings are supported by many studies that report that increased progesterone levels have significantly reduced pregnancy rates [26,27,28]. Therefore, this finding might explain why pregnancy outcomes are disrupted in patients with high serum progesterone on the day of hCG administration.

Progesterone levels on the day of administration of hCG > 1 ng/mL reduced HOXA10 expression so that which indicates that there has been a disruption in the reception of endometrium during the implantation period. Furthermore, freezing all of embryos could be the solution if P4 levels after the stimulation procedure are found to be >1 ng/mL and then transferred in the next cycle of menstruation after determining that the effect of stimulation is already at a minimal level.

## Figures and Tables

**Figure 1 biomedicines-07-00083-f001:**
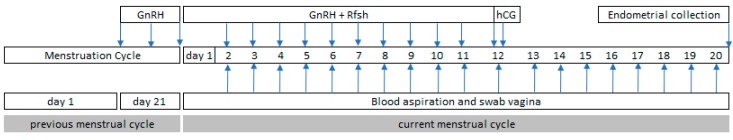
Controlled ovarium hyperstimulation (COH) procedure.

**Figure 2 biomedicines-07-00083-f002:**
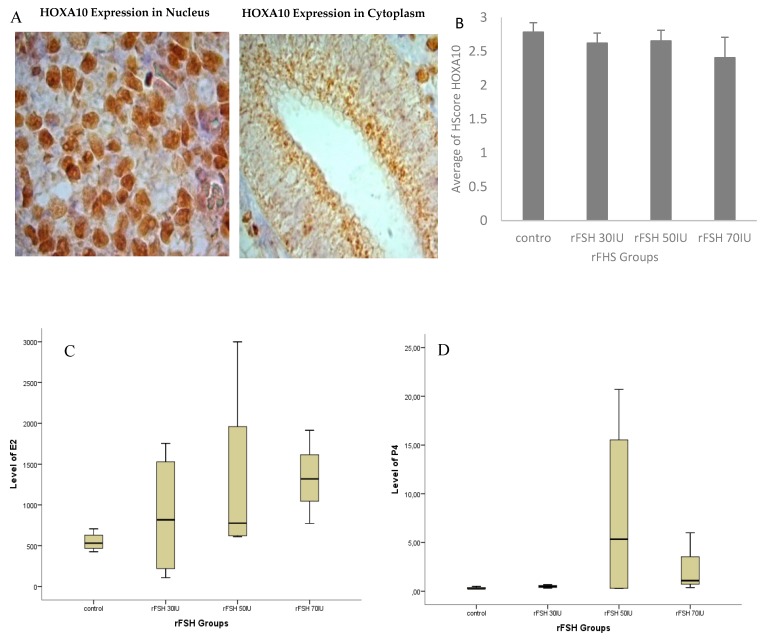
Immunohistochemical localization of homeobox A10 (HOXA10) protein in *Macaca nemestrina* endometrium. Localization of HOXA10 expression in cytoplasmic gland epithelium cells and nucleated stroma cells in 1000× magnification (**A**). Using ANOVA Average of Hscore HOXA10 in nucleated stroma cells found not statistically significant within groups (*p*-value, 0.126) (**B**); levels of E2 (**C**) and P4 (**D**) analyzed using the Kruskal–Wallis test and found not statistically significant within groups (*p*-value E2, 0.205; *p*-value P4, 0.146).

**Figure 3 biomedicines-07-00083-f003:**
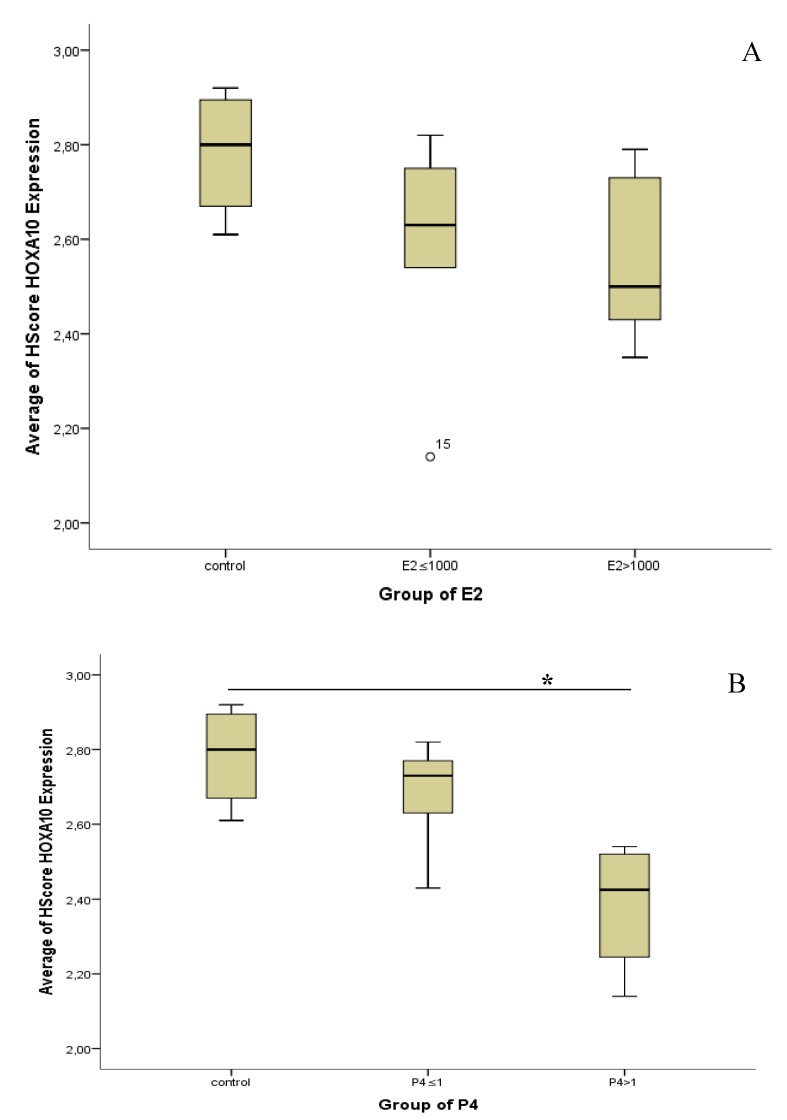
HOXA10 and steroid level on the day of hCG administration. The Kruskal–Wallis test was used, and no statistically significant differences were found in the average of the Hscore for HOXA10 within E2 groups (*p*-value, 0.190) (**A**). Statistically significant differences were found in the average of the Hscore for HOXA10 within P4 groups (*p*-value, 0.022), where the significance was found in the P4 > 1 group with the natural group (*p*-value, 0.029) (**B**).

**Figure 4 biomedicines-07-00083-f004:**
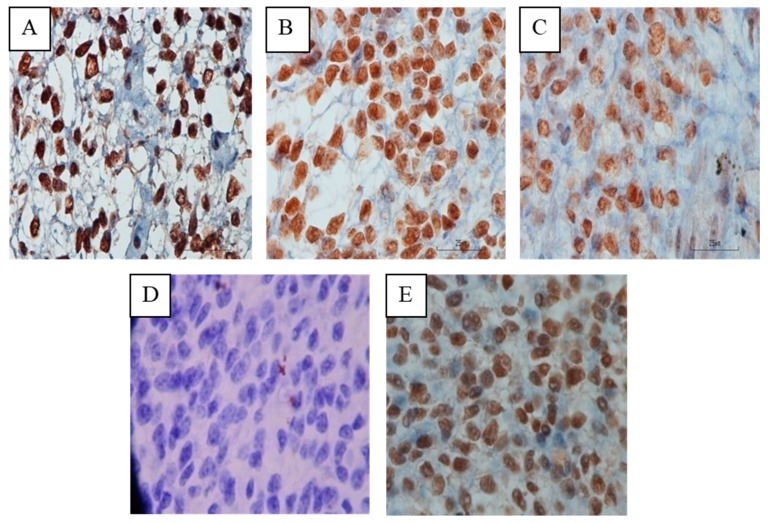
Immunohistochemistry HOXA10 in *Macaca nemestrina* endometrial tissue. The natural cycle (**A**); the stimulated cycle with P4 < 1 ng/mL (**B**); the stimulated cycle with P4 > 1 ng/mL (**C**); negative control (**D**) and positive control (**E**). Magnification 1000×.

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
