# Peer review of "Increased Progesterone on the Day of Administration of hCG in Controlled Ovarian Hyperstimulation Affects the Expression of HOXA10 in Primates’ Endometrial Receptivity"

_biomedicines, 2019, doi:10.3390/biomedicines7040083_

Round 1

Reviewer 1 Report

The manuscript by Nurhuda et al., evaluates the effect of progesterone (>1ng/ml) on endometrial receptivity (specifically looking at HOXA10 expression). Overall, this studies addresses an important issue. Listed below are my comments for the authors-

1) Material and methods- Include a flow chart describing the controlled ovarian hyperstimulation procedure and different groups for better understanding. 

2) Results- In the provided pdf, some of the images are cut off at the bottom so it is hard to see the all of the histology.

3) Immunohistochemistry is a more semi-quantitative method rather than  quantitative method. If possible, authors should include RT-qPCR for HOXA10 levels. 

4) Authors should revamp the discussion.   

Author Response

Notes to reviewer:

We have Included a flow chart describing the controlled ovarian hyperstimulation procedure in our menuscript revision. We have improved some of the images of the histology.  It was impossible to include RT-qPCR for HOXA10 levels, because we use paraffinized  tissue. We have revamped our discussion in our manuscript revision.

Reviewer 2 Report

The authors present a paper addressing the increase of progesterone and expression of HOXA10 in primates endometrium after administration of hCG in controlled ovarian hyperstimulation. The paper is very confusing and doesn't read well mainly because of English language. The title is confusing and should be rearranged. Is the increase in progesterone that leads to increased expression of HOXA10? If so, how do the authors proof that?

Also, I have some concerns in the use of the animal model. What is the name of the species? The authors refer only to genus.  Why did the authors sacrificed the animals? To obtain sera and tissue biopsy of a primate there's no need to sacrifice the animal.

Author Response

1. The increase in progesterone leads to increased expression of HOXA10 as we describes in the discussion.

2. The name of  the species is Macaca nemestrina. Macaca is the name of genus.

3. We  sacrificed the animals because other organ was isolated such as colon, liver, and spleen for other research.